# Curcumin Alleviates LPS-Induced Oxidative Stress, Inflammation and Apoptosis in Bovine Mammary Epithelial Cells via the NFE2L2 Signaling Pathway

**DOI:** 10.3390/toxins13030208

**Published:** 2021-03-12

**Authors:** Ruihua Li, Hengtong Fang, Jinglin Shen, Yongcheng Jin, Yun Zhao, Rui Wang, Yurong Fu, Yue Tian, Hao Yu, Jing Zhang

**Affiliations:** Key Laboratory of Zoonosis Research, Ministry of Education, Department of Animal Science, College of Animal Sciences, Jilin University, Changchun 130062, China; lirh18@mails.jlu.edu.cn (R.L.); fanght@jlu.edu.cn (H.F.); shenjl@jlu.edu.cn (J.S.); ycjin@jlu.edu.cn (Y.J.); zhao_yun@jlu.edu.cn (Y.Z.); ruiwang18@mails.jlu.edu.cn (R.W.); fuyr20@mails.jlu.edu.cn (Y.F.); letian20@mails.jlu.edu.cn (Y.T.)

**Keywords:** MAC-T cells, oxidative stress, inflammation, apoptosis, curcumin, LPS

## Abstract

Lipopolysaccharide (LPS) is an endotoxin, which may cause immune response and inflammation of bovine mammary glands. Mastitis impairs animal health and results in economic loss. Curcumin (CUR) is a naturally occurring diketone compound, which has attracted widespread attention as a potential anti-inflammatory antioxidant. The purpose of this study is to investigate whether CUR can reduce the damage of bovine mammary epithelial cells (MAC-T) induced by LPS and its underlying molecular mechanism. The MAC-T cell line was treated with different concentrations of LPS and CUR for 24 h. The results showed that CUR rescued the decrease of MAC-T cell viability and cell damage induced by LPS. At the same time, 10 µM CUR and 100 µg/mL LPS were used to treat the cells in the follow-up study. The results showed CUR treatment reduced the accumulation of reactive oxygen species (ROS), the expression of inflammatory cytokines (tumor necrosis factor-a (TNF-α), interleukin-8 (IL-8), IL-6 and IL-1β) and the rate of apoptosis induced by LPS. These effects were associated with the activation of the nuclear factor E2-related factor 2 (NFE2L2)-antioxidant response element (ARE) pathway coupled with inactivation of the nuclear factor-κB (NF-κB) inflammatory and caspase/Bcl2 apoptotic pathways.

## 1. Introduction

Bovine mastitis is usually caused by an inflammatory response caused by pathogenic bacteria in the bovine mammary glands, and occasionally by mechanical or chemical damage [1]. Most mastitis occurs as a low-grade infection, a subclinical state, which affects 10–15% cows, increasing milk leucocyte content, reducing milk production and increasing milk bacterial content. These all contribute to reduced milk value as a food and in monetary terms [2]. Bovine mammary epithelial cells participate in the mammary gland’s first line of defense against invading pathogens and play an important role in the initiation of infection [3]. Antibiotic treatment has been a commonly used and effective method for this disease [4]. However, with the public’s concern that antibiotic-resistant bacteria will develop and the bacteria will be transferred to humans, research on low-cost, safe and effective pharmaceutical preparations is still very necessary. At present, new mastitis therapies have gradually entered the public’s field of vision, such as injecting nanoparticles with antibacterial drugs [5], encouraging farmers to adopt homeopathy [6], and feeding antibacterial and anti-inflammatory Chinese herbal medicines [7].

Lipopolysaccharide (LPS) is the main component of the cell wall of Gram-negative bacteria, which can cause immune response and inflammation in bovine mammary epithelial cells [8]. Because there are a large number of Gram-negative bacteria in the intestines and rumen of dairy cows, they virtually constitute a huge reservoir of endotoxin [9]. In high-yielding dairy cows, it has been found that mastitis occurs frequently in dairy cows during the early and peak periods of lactation, because breeders always feed high-concentration feeds in order to obtain better dairy products [10]. On the one hand, this feeding method will lead to an abnormal decrease in rumen pH and impair the integrity of the rumen epithelium, which will cause the free LPS in the rumen to enter the portal circulation and further cause systemic inflammation [11]. On the other hand, the liver is an important organ for the body to remove LPS. However, too much LPS can damage liver function and enter the circulatory system through capillaries, causing inflammation of mastitis [12]. The study of Zhang, Lili, et al. found that *Staphylococcus aureus* phage can inhibit LPS-induced inflammation of bovine mammary epithelial cells (MAC-T) [13]. Ma, Xiao, et al. found that oligomeric proanthocyanidins play an important role in inhibiting the inflammation process of MAC-T cells induced by LPS [14]. Therefore, studying the effect of LPS on bovine mammary epithelial cells can improve our understanding of the prevention and treatment of bovine mastitis infection.

Curcumin (CUR) has been widely and safely consumed as a natural food coloring for hundreds of years, and there have been a large number of studies showing its antioxidant, anticancer and anti-inflammatory properties [15,16]. Curcumin can not only effectively remove reactive oxygen species, but also activate antioxidant response elements to inhibit oxidative stress induced by reactive oxygen species [17]. The nuclear factor E2-related factor 2 (NFE2L2)-antioxidant response element (ARE) signaling pathway plays an important role in mediating the cytoprotective response to oxidative stress, and curcumin is one of the activators of this pathway [18,19]. Using curcumin preparations with enhanced bioavailability to patients with breast cancer, it has been observed that it can significantly improve the quality of life of these patients and inhibit systemic inflammation [20]. In addition, previous studies have shown that CUR protects against LPS-induced endotoxemia by blocking oxidative stress, cytokine production and polymorphonuclear cell infiltration [21,22]. These research results fully demonstrated the anti-inflammatory and antioxidant activity of CUR, so we guessed whether it can inhibit the damage of LPS to bovine mammary cells, so as to provide a theoretical basis for the prevention and treatment of bovine mastitis in animal husbandry.

## 2. Results

### 2.1. Curcumin (CUR) Rescued the Decrease of MAC-T Cell Viability and Cell Damage Induced by Lipopolysaccharide (LPS)

We first examined the effect of CUR and LPS on breast cancer cell proliferation. Cell viability was used as a measure of cell proliferation. As shown in Figure 1A, Low-dose CUR (5, 10 µM) treatment of bovine mammary epithelial cells for 24 h can significantly improve cell viability. The viability of MAC-T cells exposed to LPS showed a concentration-dependent response. As the concentration of LPS increased, the viability of MAC-T cells decreased significantly (Figure 1B). However, 5 or 10 µM CUR can rescue the cell viability decline caused by LPS, and 10 µM CUR can significantly increase the cell viability decline caused by 100 µg/mL LPS (Figure 3A). Lactate dehydrogenase (LDH) is a relatively stable enzyme that exists in the cytoplasm of all cells. When the cell membrane is damaged, it will be quickly released into the cell culture medium. Therefore, the activity of LDH in the cell culture medium is tested to assess the integrity of the cell membrane. This reflects the degree of cell damage. As shown in Figure 1C, 10 µM curcumin significantly alleviated the cell membrane damage caused by 100 µg/mL LPS on MAC-T cells.

### 2.2. CUR Prevents LPS-Induced Oxidative Stress in MAC-T Cells

Compared with the control group, LPS treatment significantly increased the level of reactive oxygen species (ROS) in MAC-T cells (Figure 2A,B) and the content of malondialdehyde (MDA) after 24 h (Figure 2C), while it significantly reduced the activities of total superoxide dismutase (T-SOD) (Figure 2D), total antioxidant capacity (T-AOC) (Figure 2E) and glutathione (GSH) (Figure 2F). However, compared with the LPS treatment group alone, the co-treatment of CUR and LPS resulted in a significant reduction in the production of ROS and MDA caused by LPS, and significantly increased the activities of T-SOD, T-AOC and GSH (Figure 2D–F).

### 2.3. CUR Activated the Nuclear Factor E2-Related Factor 2 (NFE2L2)-Antioxidant Response Element (ARE) Pathway

As shown in Figure 3A, compared with the control cultures, LPS treatment reduced the protein levels of heme oxygenase 1 (HMOX1) and nuclear factor E2-related factor 2 (NFE2L2). However, in the presence or absence of LPS, CUR significantly increased the levels of these two proteins in MAC-T cells. In addition, the real-time polymerase chain reaction (RT-PCR) test results showed that, consistent with the Western blotting results, LPS reduced the mRNA abundance of NFE2L2 pathway downstream genes HMOX1 and NAD(P)H:quinone oxidoreductase 1 (NQO1), while compared with the LPS treatment group, CUR increased the NFE2L2 pathway gene mRNA abundance (Figure 3B–D).

### 2.4. CUR Prevents LPS-Induced Decrease in Mitochondrial Membrane Potential (ΔΨm, MMP)

Next, we use the methyl benzimidazole and dichloromethane iodide (JC-1) fluorescent probe to measure the mitochondrial membrane potential changes of MAC-T under different treatments. It can be seen from Figure 4A, B that the MMP value of the LPS + CUR group was significantly higher than that of the LPS group. This indicates that CUR can reverse the damage of cell mitochondria caused by LPS by promoting energy metabolism.

### 2.5. CUR Rescues the Apoptosis Caused by LPS Treatment of MAC-T Cells

In order to determine the effects of LPS and CUR on MAC-T cell apoptosis, we used three methods for detection. First, flow cytometry was used to detect the apoptosis rate of MAC-T cells. As shown in Figure 5A, LPS treatment resulted in a significant increase in the rate of apoptosis after 24 h, while CUR can significantly rescue LPS-induced apoptosis. Then the relative expression levels of BCL2-Associated X (Bax) and B-cell lymphoma-2 (Bcl-2) genes were detected by real-time quantitative PCR. The results showed that LPS significantly up-regulated the expression of Bax and significantly down-regulated the expression of Bcl-2 (Figure 5B–D). The combined CUR treatment has the opposite effect. Finally, we studied the effects of LPS and CUR on apoptosis-related proteins by Western blotting. As shown in Figure 5E, LPS treatment significantly increased the protein content of caspase-3 and caspase-9 in MAC-T cells, while the LPS+CUR group could significantly resist this effect. These results indicate that LPS treatment has a pro-apoptotic effect on MAC-T cells.

### 2.6. CUR Rescued LPS-Elicited Nuclear Factor Kappa-B (NF-κB) Signaling Pathway Activity

Culturing with LPS significantly increased the expression of nuclear factor kappa-B p65 (NF-κB p65) and NF-κB p50. In contrast, compared with the LPS-treated group, the expression of NF-κB p65 and NF-κB p50 decreased in the CUR + LPS-treated group (Figure 6A,B). Similarly, the expression of the NF-κB pathway target genes interleukin-8 (IL-8) and interleukin-1β (IL-1β) was also increased in the LPS-treated group (Figure 6C,D), and the use of enzyme-linked immunosorbent reaction (ELISA) kits found that the levels of interleukin-6 (IL-6) and tumor necrosis factor-a (TNF-a) protein increased significantly, while CUR attenuated the LPS-induced expression of inflammatory cytokines (Figure 6E,F).

## 3. Discussion

Mastitis affects the health and milk production of dairy cows, leading to economic losses in the dairy industry. LPS is an extremely important cell wall component of Gram-negative bacteria, which can cause mastitis in dairy cows [23]. The results of this study showed that treating MAC-T cells with 100 µg/mL LPS for 24 h not only caused a significant decrease in cell viability, but also damaged the integrity of the cell membrane. Low-dose CUR (5, 10 µM) treatment of MAC-T cells for 24 h can significantly improve the cell viability of MAC-T cells and resist the damage caused by LPS (Figure 1). We believe this is related to CUR’s promotion of cell growth and metabolism. For example, in the study of Mehta and Shikhar et al., CUR increased chondrocytes viability of the cell [24]. In addition, CUR can significantly improve the decrease in cell viability and cell damage caused by LPS. This protective effect suggests that it can be used as a potential drug for the treatment of bovine mastitis.

In the pathogenesis of mastitis, the release of ROS is an important part of the inflammatory response [25]. In the transitional period, bovine mammary cells will undergo a strong metabolism and further increase the release of ROS [26]. On the one hand, the excessive ROS is closely related to cell inflammation and apoptosis [27,28]. On the other hand, the excessive ROS and the anti-oxidant defense system are unbalanced, which leads to the oxidation of cell lipids, proteins and DNA, causing substantial damage to nearby tissues. CUR has the antioxidant ability to directly eliminate ROS and stimulate the endogenous cellular defense system [29]. In this study, 10 µM CUR significantly reduced the accumulation of ROS induced by LPS in MAC-T cells (Figure 2A,B). This effect is consistent with the research results of Xiang and Biao, et al. [30]. In addition, CUR significantly reduced the increase in MDA content in cells induced by LPS (Figure 2C). For LPS-induced decrease in T-SOD activity, decrease in total antioxidant capacity and decrease in GSH activity, these effects were significantly inhibited by CUR (Figure 2D–F). Therefore, the increase of SOD and GSH activities under oxidative stress conditions highlights the positive effects of CUR on the cellular antioxidant defense system. The decrease of ROS and MDA concentration also indicates that CUR has a positive effect on the activity of scavenging free radicals.

The NFE2L2-ARE signaling pathway plays a key role in maintaining cell redox balance [31]. Yang, Chenhui et al. found that CUR can up-regulate the expression of transcription factors NFE2L2 and HMOX1 to protect the rat brain from focal ischemia [32]. Previous studies have shown that CUR activates the expression of NFE2L2 and protects cells against oxidative stress, indicating that NFE2L2 is the key to the up-regulation of antioxidant enzymes [19]. Therefore, CUR attenuates the inhibitory effect of LPS on the protein expression of NFE2L2 and HMOX1 and the mRNA transcription level of NQO1, which is consistent with previous findings. In the result of Figure 3, the content of NFE2L2 protein and HMOX1 protein in MAC-T cells was significantly increased by 50% in the CUR-treated group compared with the control group. Not only does the treatment of MAC-T cells with CUR significantly activate the NFE2L2-ARE pathway, it also significantly reduces the inhibitory effect of LPS on this pathway. In the results shown in Figure 4, CUR increases the mitochondrial membrane potential of cells, indicating that it promotes intracellular energy metabolism.

A large number of studies have shown that LPS can induce cell oxidative stress and lead to apoptosis. In Figure 5A, flow cytometry results show that LPS significantly induces MAC-T cell apoptosis, while CUR treatment can reduce the apoptosis rate by about 70%, which shows that CUR protects MAC-T cells effect. As the starting caspase component of the apoptotic body complex, caspase 9 can be activated by ROS [33]. The caspase 9 can activate the downstream caspase 3 to initiate the caspase cascade, which leads to cell apoptosis [34]. In addition, Bcl-2 is an anti-apoptotic gene that can inhibit the generation of oxygen free radicals and stabilize the mitochondrial membrane potential, while Bax plays a role in promoting cell apoptosis [35]. In this study, CUR inhibited LPS-induced apoptosis, and as shown by the change in the ratio of Bax/Bcl-2, inhibited the mitochondrial apoptotic pathway and reversed the caspase apoptotic pathway, this indicates that CUR can rescue MAC-T apoptosis induced by LPS.

Oxidative stress and enhanced systemic disease states may be common factors that lead to transitional dairy cattle metabolic and infectious diseases [36]. ROS can activate the redox-sensitive transcription factor NF-κB, thereby increasing the expression of pro-inflammatory cytokines. Excessive activation of the NF-κB inflammatory pathway may lead to a chronic increase in the expression of inflammatory cytokines. Pro-inflammatory cytokines play a vital role in the development of inflammation, such as TNF-α, IL-1β, IL-8, and IL-6 [37]. These are well-known pro-inflammatory cytokines and chemokines, which involve many types of inflammation including mastitis [38,39,40,41]. In the results in Figure 7, LPS activated the NF-κB pathway and induced the increase of inflammatory cytokines, which indicated that LPS induced the occurrence of inflammation in MAC-T cells, and the anti-inflammatory effect shown by CUR indicated that it has the ability to treat mastitis.

It is worth noting that LPS is the main surface membrane component of almost all Gram-negative bacteria [42]. According to the extracted bacterial strains, it can be divided into *Escherichia coli* type, *Salmonella typhosa* type and *Pseudomonas aeruginosa* 10. According to the serotype (not a separate species), it can be divided into *Escherichia coli* 0111: B4 serotype, *Escherichia coli* 026: B6 serotype, *Salmonella enterica* subsp. *enterica* serotype Enteritidis, *Salmonella enterica* subsp. *enterica* serotype Typhimurium and *Salmonella enterica* subsp. *enterica* serotype Minnesota etc. [43]. The biological activity of LPS depends not only on the bacterial species from which it is derived, but also on the serotype of the bacterial species. For example, according to a report by Mikołajczyk, when treating porcine dorsal root ganglion primary cells with different serotypes of LPS, although the average number of neurons was not changed, the proportion of substance P (SP)-positive neurons after treatment with LPS *S*. Enteritidis was significantly higher than that of the control, while the proportion of these cells treated with LPS *S*. Minnesota and LPS *S*. Typhimurium decreased significantly [44]. The one from *Escherichia coli* serotype 055: B5 is used to induce myocarditis [45]. LPS extracted from *Salmonella* Typhi can induce endotoxemia and microvascular thrombosis in mice [46]. LPS from serotype *Salmonella enteritidis* is used for the role of inflammation in neuropathic pain and aneurysm progression [47,48]. Treatment of MAC-T cells with LPS from *Escherichia coli* J5 (Rc mutant) for 24 h can induce an increase in the expression of Toll-like receptor (TLR) 4 and downstream TLR signaling molecules [49]. In Lili’s study, treatment of MAC-T cells with 1 µg/mL of LPS (LPS type not specified) for 5 h induced a significant increase in cellular inflammatory factors [13]. According to Wang’s report, MAC-T cells treated with LPS (100 µg/mL, *Escherichia coli* 055: B5) for 12 h induced apoptosis and activated the NF-κB inflammatory pathway [50]. The type of LPS used in this study was *Escherichia coli* serotype 0111: B4 purchased from Sigma-Aldrich. Based on the effect of 100 µg/mL LPS on MAC-T cells for 24 h, it can be considered that it has successfully induced the inflammation of MAC-T cells, and it can be used as an inflammation model for studying bovine mastitis. This is consistent with Fan’s research results [51].

Preventing mastitis is better than controlling mastitis. Preventing mastitis is better than controlling mastitis. At present, Inmufort Bov (LPS from *Ochrobactrum intermedium*) has been found to effectively reduce the number of somatic cells in milk, significantly reduce the incidence of subclinical mastitis, and reduce the intensity of clinical signs of mastitis. Field studies have confirmed its immunostimulatory activity in cattle (mainly dairy cows). Not only can it reduce the use of antibiotics to treat mastitis, but it can also significantly increase the effectiveness of this treatment [52].

At the same time, the poor bioavailability of CUR is also worth noting. The chemical structure of CUR includes two phenols, which makes it difficult to dissolve in water and more soluble in organic solvents [53]. There are still many difficulties in taking turmeric by mouth to achieve good results. There are two main reasons for its low bioavailability. Firstly, the poor water solubility of turmeric causes poor absorption by the organism. Secondly, its metabolism in the organism intestine and liver is very fast [54]. At present, the use of polymer synthesis technology to prepare amphiphilic polycurcumin block copolymers with CUR as a comonomer has overcome the shortcomings of CUR in water solubility and poor stability [55,56,57]. Although low-dose CUR can protect MAC-T cells from LPS damage in our research results, if you want to use it in animal husbandry production A new in vivo experiment is needed to determine the most suitable CUR dosage.

## 4. Conclusions

The results of this study provide important evidence for the potential cytoprotective effect of CUR on LPS-induced damage in MAC-T cells. The mechanism of action includes reducing ROS production, maintaining intracellular redox balance, activating the NFE2L2 pathway, but inactivating NF-κB inflammation and caspase/Bcl2 apoptosis pathways. It provides an in vitro test basis for the application of CUR in the prevention and treatment of bovine mastitis in animal husbandry.

## 5. Materials and Methods

### 5.1. Chemicals and Reagents

The dimethyl sulfoxide (DMSO), lipopolysaccharide (*Escherichia coli* 0111: B6) and CUR were purchased from Sigma (Sigma, St Louis, MO, USA). Antimycotics, fetal bovine serum (FBS) and Dulbecco’s modified Eagle’s high glucose medium were purchased from HyClone (HyClone, Logan, UT, USA). Cell Counting Kit-8 (CCK-8) was obtained from Dojindo (Dojindo Laboratories, Kumamoto, Japan). The malondialdehyde (MDA) assay kit, total superoxide dismutase (T-SOD) assay kit, total antioxidant capacity (T-AOC) assay kit and reduced glutathione (GSH) assay kit were purchased from Jiancheng (Jiancheng, Nanjing, China). 2′,7′-Dichlorodihydrofluorescein diacetate (DCFH-DA), annexin V and propidium iodide (PI) were purchased from Beyotime (Beyotime, Shanghai, China). For Western blot analysis, antibodies against HMOX1, NFE2L2, caspase-3 and β-actin were purchased from Abcam (Abcam, Cambridge, MA, USA), and antibody against caspase-9 was purchased from Affinity Biosciences (Affinity, Cincinnati, OH, USA). The ELISA kits used in the experiment were purchased from Sigma (Sigma, St Louis, USA).

### 5.2. Cell Culture

Bovine MAC-T mammary epithelial cells were kindly provided by H.G. Lee (Konkuk University, Korea). The MAC-T cell line was cultured in Dulbecco’s modified Eagle’s high glucose medium supplemented with 10% FBS, 100 UI/mL penicillin, 100 µg/mL streptomycin, 1 μg/mL hydrocortisone and 5 μg/mL insulin at 37 °C in a humidified atmosphere with 5% CO_2_. Cultures were tested periodically and confirmed to be mycoplasma-free.

### 5.3. Treatment Methods for Cells

CUR was dissolved in DMSO at a concentration of 50 mM for storage and diluted to specific concentrations in cell culture medium for cell treatments. The CUR stock solution was diluted in cell culture medium to concentrations of 2.5, 5, 10, 20, 40 µM, and the same volume of DMSO was added to the control groups. The final concentration of DMSO in the treatment solutions prepared above was less than 0.1% (*v/v*). LPS is dissolved with cell culture medium to a specific concentration (20, 40, 80, 100, 120, 200, 240, 300, 400 µg/mL) for cell treatment. In subsequent experiments, the concentration of CUR was 10 µM and the concentration of LPS was 100 µg/mL to treat MAC-T cells for 24 h.

### 5.4. Cell Viability Assay

Cell viability was determined by the Cell Counting Kit-8 (CCK-8; Dojindo, Tokyo, Japan). Cells were seeded into 96-well plates at a density of 0.2 × 10^4^ cells per well, and after CUR or/and LPS treatment of MAC-T cells for 24 h, 10 μL CCK-8 reagent was added to each well. Plates were incubated for 1 h at 37 °C, and the absorbance value (OD) of each well was measured at 450 nm according to the manufacturer’s instructions.

### 5.5. Staining with Annexin V and Propidium Iodide (PI)

After treatments, cells were collected and stained with Annexin V-Fluorescein isothiocyanate isomer (FITC) reagent and PI followed by analysis by flow cytometry. The percentage of dead cells was quantified using FlowJo 10.5 software (Ashland, OR, USA: Becton, Dickinson and Company; 2019).

### 5.6. Fluorescence Measurements of Intracellular Oxidants

DCFH-DA was dissolved in dimethyl sulfoxide (DMSO) and stored at −20 °C in a 10 mM stock solution. MAC-T cells were treated with CUR and LPS for 24 h, incubated with 10 μM DCFH-DA for 30 min, washed 3 times with PBS, and immediately observed and imaged using a fluorescence microscope; three photos were required for each treatment group.

### 5.7. Measurement of Malondialdehyde (MDA), Total Superoxide Dismutase (T-SOD), Total Antioxidant Capacity (T-AOC) and Glutathione (GSH) Content

The cellular MDA, T-SOD, T-AOC and GSH contents were determined using commercial kits. The MDA level is expressed as nmol/mg protein in relation to the cellular protein concentration. The GSH level is expressed as µmol/gprot protein in relation to the cellular protein concentration.

### 5.8. RNA Isolation and Quantitative Real-Time Polymerase Chain Reaction (PCR)

Genes obtained β-actin, HMOX1, NFE2L2, NQO1, Bax, Bcl-2, NF-κB P65, NF-κB P50, IL-6 and IL-1β Coding sequences (CDS) in NCBI, and placed the CDS region in the online Primer 5 The software designs the upstream and downstream primer sequences (Real-time PCR primer sequence list); use the Primer Basic Local Alignment Search Tool (BLAST) on National Center for Biotechnology Information (NCBI) for primer verification. The primer sequence synthesis was completed in Suzhou Jinweizhi Company, and β-actin was used as the internal reference. Add 10.0 μL of synergy brands (SYBR) Green, 0.4 μL of upstream primers, 0.4 μL of downstream primers, 7.2 μL of ddH2O, and 2 μg (2.0 μL) of cDNA to an eight-connected tube with 20 μL fluorescence quantitative reaction system. This was mixed well and centrifuged briefly to make the mixture sink to the bottom of the tube. Then the prepared eight-strip tube was put into the real-time fluorescent quantitative PCR instrument, and it was melted at 94 °C, 30 s pre-denaturation, 94 °C, 5 s, 60 °C, 15 s annealing temperature, 72 °C, 15 s extension. The number of cycles was generally 40, and the number of cycles could be adjusted according to different primer reactions. Fluorescence quantitative results were calculated using the 2^−ΔΔct^ method to calculate the relative mRNA levels of genes. The primers used in the qPCR analyses are listed in Table 1.

### 5.9. Western Blot Analysis

The total protein of the cells after the culture was extracted according to the test instructions, and the concentration of the extracted protein was determined by the bicinchoninic acid (BCA) method. Mix the protein sample with the loading buffer and heat it in a water bath at 95 °C for 10 min. Configure a suitable sodium dodecyl sulfate polyacrylamide (SDS-PAGE) protein separation gel according to the size of the target protein. Add 30 µg protein sample to each sample in the air, and transfer it to 0.45 µm ethyl acetate membrane according to the wet transfer method after electrophoresis is completed. The membranes were washed one time with tris buffered saline tween (TBST) buffer and incubated with a suitable primary rabbit antibody (1:1000) specific for HMOX1, NFE2L2, caspase-3 or caspase-9 at 4 °C overnight. After washing four times with TBST, the immunoblotted membranes were incubated with a horseradish peroxidase-labeled goat antirabbit immunoglobulin G (IgG)-conjugated secondary antibody for 2 h at room temperature. Finally, using a Pierce Emitter Coupled Logic (ECL) substrate, protein bands were imaged on a chemiluminescence imaging analyzer.

### 5.10. Statistical Analysis

All experiments were performed independently at least three times, and the data are expressed as the mean ± standard error of the mean (SEM). GraphPad PRISM software (Windows 5.02; GraphPad Software, Inc.) was used to test the significance of the data by one-way analysis of variance, and *p*  <  0.05 was deemed a statistically significant difference.

## Figures and Tables

**Figure 1 toxins-13-00208-f001:**
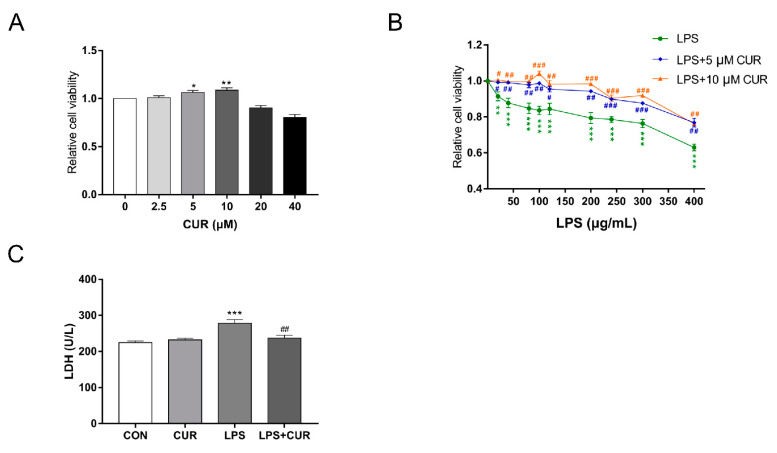
Curcumin (CUR) rescued the decrease of MAC-T cell viability and cell damage induced by lipopolysaccharide (LPS). (**A**) The cell viability of MAC-T cells 24 h after CUR treatment. (**B**) The cell viability of MAC-T cells treated with LPS and LPS combined with CUR (5, 10 µM) for 24 h. (**C**) The lactate dehydrogenase content of MAC-T cells treated with turmeric or LPS for 24 h. The data of the untreated group were used to normalize the data of each treatment group. Data are means ± standard errors of the mean (SEM) of 3 independent experiments. * *p* < 0.05; ** *p* < 0.01; *** *p* < 0.001, compared to CON (control). ^#^
*p* < 0.05; ^##^
*p* < 0.01; ^###^
*p* < 0.001, compared to LPS group.

**Figure 2 toxins-13-00208-f002:**
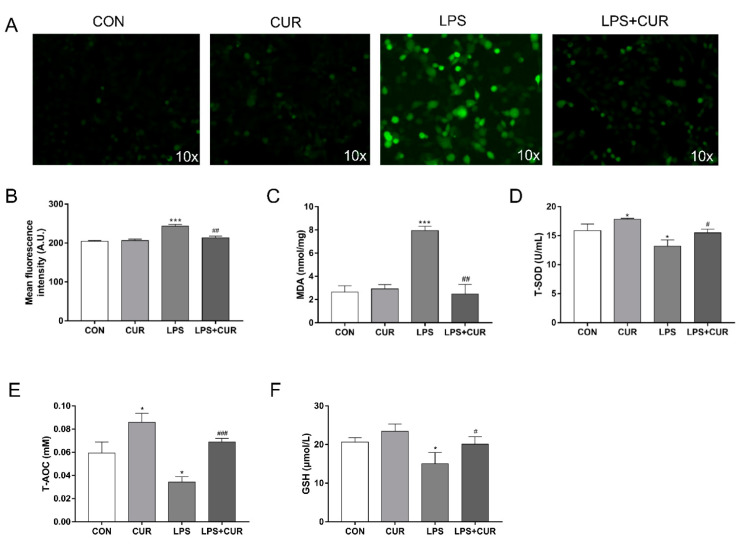
CUR prevents LPS-induced oxidative stress in MAC-T cells. MAC-T cells were treated with CUR (10 μM) and/or LPS (100 μg/mL) for 24 h. (**A**) After incubation with 5 μM 2′,7′-dichlorodihydrofluorescein diacetate (DCFH-DA), cells were washed and examined by fluorescence microscopy (scale bar represents 200 µm). Representative images from three independent experiments are shown. (**B**) The results of reactive oxygen species (ROS) upregulation in breast cancer cells were determined by fluorescence analysis using ImageJ software. (**C**) Malondialdehyde (MDA) content in MAC-T cells. (**D**) Total superoxide dismutase (T-SOD) activity in MAC-T cells. (**E**) Total antioxidant capacity (T-AOC) activity in MAC-T cells. (**F**) Glutathione (GSH) activity in MAC-T cells. The data of the untreated group were used to normalize the data of each treatment group. Data are means ± SEM of 3 independent experiments. * *p* < 0.05; *** *p* < 0.001, compared to CON (control). ^#^
*p* < 0.05; ^##^
*p* < 0.01; ^###^
*p* < 0.001, compared to LPS group.

**Figure 3 toxins-13-00208-f003:**
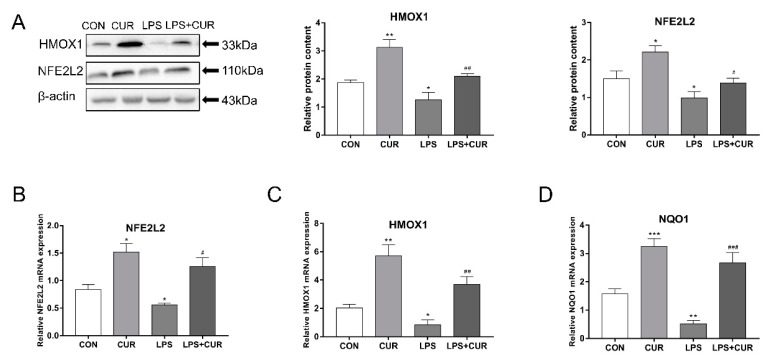
CUR activated the nuclear factor E2-related factor 2 (NFE2L2)-antioxidant response element (ARE) pathway. MAC-T cells were treated with CUR (10 μM) and/or LPS (100 μg/mL) for 24 h. (**A**) Protein levels of heme oxygenase 1 (HMOX1) and NFE2L2. NFE2L2 mRNA level in MAC-T cells. (**B**) NFE2L2 mRNA level in MAC-T cells. (**C**) HMOX1 mRNA level in MAC-T cells. (**D**) NQO1 mRNA level in MAC-T cells. The data of the untreated group were used to normalize the data of each treatment group. Data are means ± SEM of 3 independent experiments. * *p* < 0.05; ** *p* < 0.01; *** *p* < 0.001, compared to CON (control). ^#^
*p* < 0.05; ^##^
*p* < 0.01; ^###^
*p* < 0.001, compared to LPS group.

**Figure 4 toxins-13-00208-f004:**
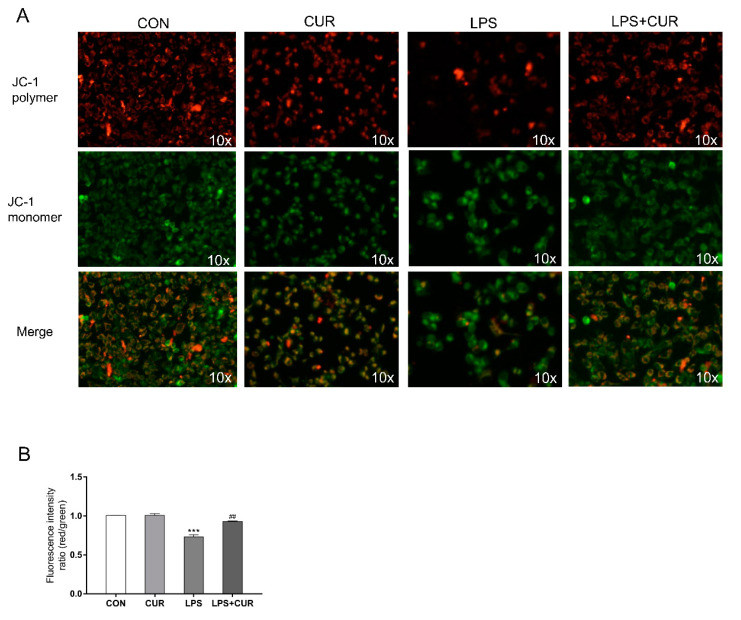
CUR prevents LPS-induced decrease in mitochondrial membrane potential (*ΔΨm,* MMP). MAC-T cells were treated with CUR (10 μM) and/or LPS (100 μg/mL) for 24 h. (**A**) Inverted fluorescence microscopy of MAC-T cells after JC-1 staining (scale bar represents 200 µm). Fluorescence analysis of MMP at 24 h using the probe JC-1. Representative images from three independent experiments are shown. (**B**) The different ratios (JC-1 polymer/JC-1 monomer) of JC-1 fluorescence for each group. The data of the untreated group were used to normalize the data of each treatment group. Data are means ± SEM of 3 independent experiments. *** *p* < 0.001, compared to CON (control). ^##^
*p* < 0.01, compared to LPS group.

**Figure 5 toxins-13-00208-f005:**
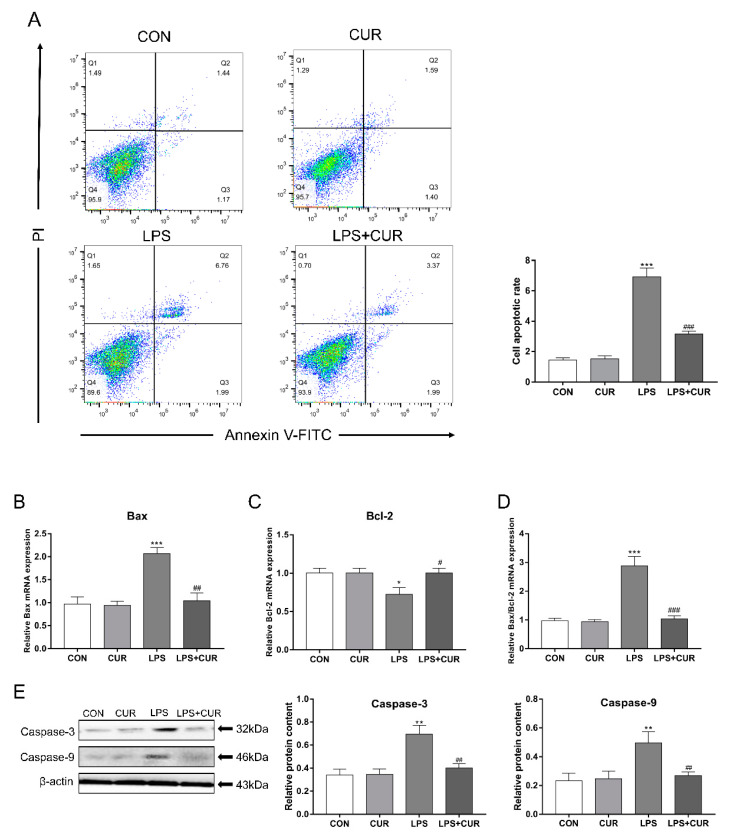
CUR rescues the apoptosis caused by LPS treatment of MAC-T cells. MAC-T cells were treated with CUR (10 μM) and/or LPS (100 μg/mL) for 24 h. (**A**) Cell death was assessed by annexin V/PI staining. (**B**) Bax mRNA level in MAC-T cells. (**C**) Bcl-2 mRNA level in MAC-T cells. (**D**) The ratio of the mRNA levels of Bax to Bcl-2. (**E**) Protein levels of caspase-3 and caspase-9. The data of the untreated group were used to normalize the data of each treatment group. Data are means ± SEM of 3 independent experiments. * *p* < 0.05; ** *p* < 0.01; *** *p* < 0.001, compared to CON (control). ^#^
*p* < 0.05; ^##^
*p* < 0.01; ^###^
*p* < 0.001, compared to LPS group.

**Figure 6 toxins-13-00208-f006:**
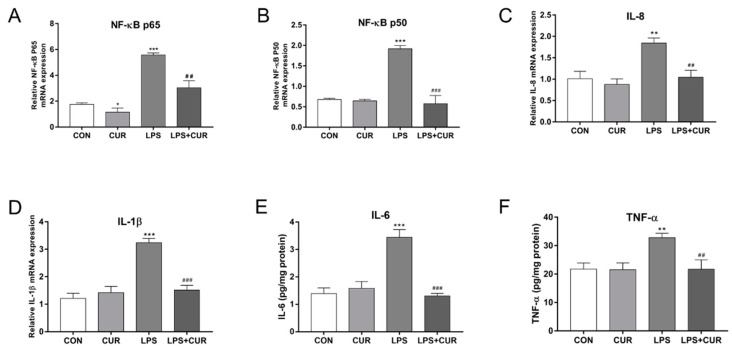
CUR rescued LPS-elicited nuclear factor kappa-B (NF-κB) signaling pathway activity. MAC-T cells were treated with CUR (10 μM) and/or LPS (100 μg/mL) for 24 h. (**A**) NF-κB p65 mRNA level in MAC-T cells. (**B**) NF-κB p50 mRNA level in MAC-T cells. (**C**) IL-8 mRNA level in MAC-T cells. (**D**) Interleukin-1β (IL-1β) mRNA level in MAC-T cells (**E**) Protein levels of IL-6 in MAC-T cells. (**F**) Protein levels of tumor necrosis factor-α (TNF-α) in MAC-T cells. The data of the untreated group were used to normalize the data of each treatment group. Data are means ± SEM of 3 independent experiments. * *p* < 0.05; ** *p* < 0.01; *** *p* < 0.001, compared to CON (control). ^##^
*p* < 0.01; ^###^
*p* < 0.001, compared to LPS group.

**Figure 7 toxins-13-00208-f007:**
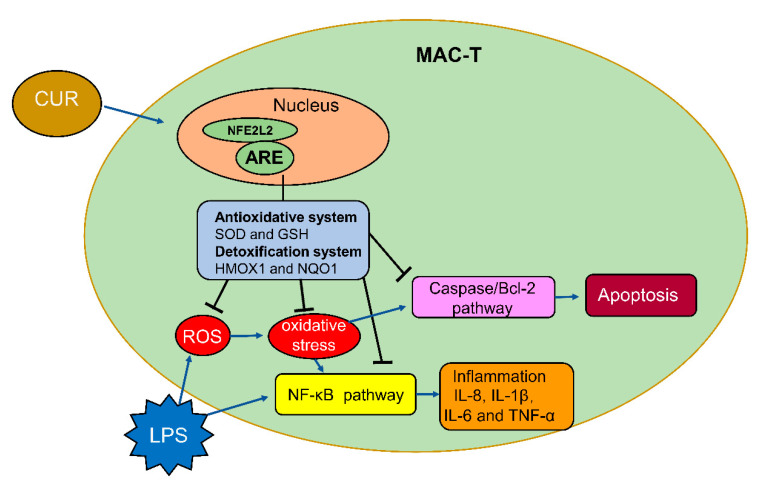
Schematic of the proposed model. CUR against LPS-induced oxidative damage in MAC-T cell via decreasing reactive oxygen species (ROS) generation, maintaining the intracellular redox balance, activating the NFE2L2 pathway, but inactivating the NF-κB inflammatory and caspase/Bcl2 apoptotic pathways.

**Table 1 toxins-13-00208-t001:** Gene name and polymerase chain reaction (PCR) primer sequences.

Gene	Gene ID	Forward Primer (5′→3′)	Reverse Primer (5′→3′)
β-actin	280979	CCCTGGAGAAGAGCTACGAG	GTAGTTTCGTGAATGCCGCAG
HMOX1	513221	TTAAGCTGGTGATGGCGTCT	GGGAGTGTAGACGGGGTTCT
NFE2L2	497024	CCCAGTCCAACCTTTGTCGT	TGGAGAGCTTTTGCCCGTAG
NQO1	519632	CACTCTGCACTTCTGTGGCT	CAGGATCTGAACTCGGGCAT
Bax	280730	GCTCTGAGCAGATCATGAAGAC	CAATTCATCTCCGATGCGCT
Bcl-2	281020	GATGACCGAGTACCTGAACC	AGAGACAGCCAGGAGAAATCA
NF-κB P65	508233	ACCTGGGGATCCAGTGTGTA	ACGGCATTCAGGTCGTAGT
NF-κB P50	616115	AAACACTGTGAGGATGGCGT	AGGCATCTGTCATTCGTGCT
IL-8	280828	ATGACTTCCAAGCTGGCTGTT	GGTTTAGGCAGACCTCGTTT
IL-1β	281251	GTCCTCCGACGAGTTTCTGT	AGAGCCTTCAGCACACATGG

## Data Availability

Data sharing not applicable.

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
