# Peer review of "Curcumin Alleviates LPS-Induced Oxidative Stress, Inflammation and Apoptosis in Bovine Mammary Epithelial Cells via the NFE2L2 Signaling Pathway"

_toxins, 2021, doi:10.3390/toxins13030208_

Round 1
Reviewer 1 Report
The purpose of this study was to evaluate the regulatory effect of CUR on the damage of LPS-induced bovine mammary epithelial cell line (MAC- 53T).
CUR has fully demonstrated its antioxidant, anti-inflammatory properties, and its potential value as an anti-hepatocellular carcinoma activity, even its poor oral bioavailability limits for its application.
- Authors observed that only the low-dose CUR (5,10µM) treatment of bovine mammary epithelial cells for 24 h can significantly improved cell viability, supporting their hypothesis.
- This phenomena is only MAC-T cell line specific?
- The poor CUR bioavailability limit also seems to be discussed in Figure 7.
Reviewer 2 Report
Manuscript ID: toxins-1119620
Title: Title: Curcumin Alleviates LPS-Induced Oxidative Stress, Inflammation and Apoptosis in Bovine Mammary Epithelial Cells via the NFE2L2 Signaling Pathway
This article can be an important consideration for some investigators, but there are some comments that need to be addressed by authors:
1) The introduction presents the literature background with some gaps in the literature. The authors should describe some sentences with additional recent literature to minimize gaps in the ‘introduction section’ of the revised manuscript.
2) Similarly to the introduction, there is a gap in the literature in the section of the discussion. The discussion presents some past data but lacks in-depth analysis and discussion of the study data.
3) Please mark in important parts of the text that your LPS is from (I think) E. coli - what serotype ... (especially in the section materials and methods). The potential readers of this paper should see an explanation of the choice of LPS ....from..... maybe E.coli.
4) Please write about LPS from different species and serotypes of bacteria in the paragraphs of your manuscript especially in the discussion. Please approach problems from a broader perspective, involving a variety of different LPS activities.
Disturbances, under the influence of LPS, depending on its origin (not only within the species but also the serotype of bacteria), may contribute to pathological processes associated with various diseases (starting with inflammation and septic shock, and ending with Parkinson’s or Alzheimer’s disease).
Even the use of a low dose of LPS in the in vitro studies involving LPS derived from three different Salmonella serotypes did not change the number of DRG neurons but affected their neurochemistry, an example being the activity of LPS from S. Enteritidis resulting in an increase in the percentage of SP-positive neurons, while LPS from S. Minnesota and LPS from S. Typhimurium exerted the contrary effect [Mikołajczyk et al., IJMS. 2018, 19(9) DOI: 10.3390/ijms19092551]. So It should be noted that the activity of even low doses of LPS on the neurons may vary and may depend not only on the species but also on the serotype of the bacteria.
5) Please detailedly discuss both the importance of LPS serotype, dose and duration time (with references ), and LPS contribution to pathological processes associated with various diseases (with references) and write about some vaccines with LPS that are used in mastitis ( for example INMUFORT BOV (OVEJERO GROUP, Bovet Drwalew; permission in Poland - of course, information of the name of the vaccine is only for the authors, not for use in the manuscript, information of the name of the vaccine is only for the authors, not for use in the manuscript).
This is particularly important in the era of LPS used for the development of anti-cancer therapies, vaccines, or immunostimulants, which are more and more commonly used.
Round 2
Reviewer 2 Report
Comments and Suggestions for Authors
Manuscript ID: toxins-1119620 - Revised Version Review
Title: Title: Curcumin Alleviates LPS-Induced Oxidative Stress, Inflammation and Apoptosis in Bovine Mammary Epithelial Cells via the NFE2L2 Signaling Pathway
Unfortunately, the authors did not fully address the comments of the reviewer. We still do not know the dose or time of LPS work. The authors refer to the work of Fan, where LPS significantly inhibited MAC-T cell proliferation in a dose- and time-dependent manner.
Unfortunately, the authors did not fully address the comments of the reviewer. We still do not know the dose or time of LPS work. The authors refer to the work of Fan, where LPS significantly inhibited MAC-T cell proliferation in a dose- and time-dependent manner.
- The introduction and the discussion still present the literature background with some gaps in the literature.
- Now we know that the authors used in this study E. coli serotype 0111: B4, but .... from......(?) dose....(?) time....(?)
- Please write more about LPS from different species and serotypes of bacteria in the paragraphs of your manuscript especially in the discussion. Please approach problems from a broader perspective, involving a variety of different LPS activities. Lipopolysaccharides have an important role in infection and survival in the host and are therefore an important virulence factor. You should take into account diverse LPS biological activity depends not only on the species but also on the serotypes (serotypes in one species).
- You should not give names of vaccines. Please do not describe different vaccines for the treatment of bovine mastitis such as the DNA/protein vaccine. You should focus on vaccines for the treatment of bovine mastitis that contain LPS.
- All abbreviations should be explained when the first time used for example, “lipopolysaccharide (LPS)”. Please definite all abbreviations.
- The authors should be familiar with the nomenclature of the microorganisms they use in the text. Please remember about Salmonella nomenclature – “Salmonella” and “S.” are italicized (of course Typhi or Typhimurium or spp. are not italicized). It is very important.
Round 3
Reviewer 2 Report
Manuscript ID: toxins-1119620 - Revised Version Review
Title: Title: Curcumin Alleviates LPS-Induced Oxidative Stress, Inflammation and Apoptosis in Bovine Mammary Epithelial Cells via the NFE2L2 Signaling Pathway
Unfortunately, the authors did not fully address the comments of the reviewer.
- First of all, we should know about the dose and the time (the dosage -100 µg/mL, and the time - 24h) from the materials and methods section and then you can write about it in the other sections like results or discussion.
- Please improve the nomenclature of the microorganisms they use in the text. Please remember about Salmonella spp. nomenclature – “Salmonella” and “S.” are italicized but Typhi or Typhimurium are capitalized and are not italicized). For named serotypes, to emphasize that they are not separate species, the serotype name is not italicized and the first letter is capitalized.
- Line 270-271 ”However, there is no publicly available commercial bovine mastitis vaccine prepared with LPS.” - Inmufort Bov (LPS from Ochrobactrum intermedium) is a stimulator of bovine mammary gland non-specific immunity. This preparation significantly reduces the incidence of subclinical mastitis, as well as reduces the intensity of mastitis clinical signs. Please remove the sentence ”However, there is no publicly available commercial bovine mastitis vaccine prepared with LPS.”
- You should take into account diverse LPS (even low doses of LPS !!!) biological activity depends not only on the species but also on the serotypes (serotypes in one species). Disturbances, under the influence of LPS, depending on its origin (not only within the species but also the serotype of bacteria), may contribute to pathological processes associated with various diseases. It should be noted that the activity of even low doses of LPS may vary and may depend not only on the species but also on the serotype of the bacteria [Mikołajczyk et al., IJMS. 2018, 19(9) DOI: 10.3390/ijms19092551].
